# Gut Microbiome-Derived Short-Chain Fatty Acids in Glomerular Protection and Modulation of Chronic Kidney Disease Progression

**DOI:** 10.3390/nu17172904

**Published:** 2025-09-08

**Authors:** Maria Szrejder, Agnieszka Piwkowska

**Affiliations:** Laboratory of Molecular and Cellular Nephrology, Mossakowski Medical Research Institute, Polish Academy of Sciences, 80-308 Gdańsk, Poland; apiwkowska@imdik.pan.pl

**Keywords:** microbiome, short-chain fatty acids, chronic kidney disease, glomerular injury

## Abstract

Chronic kidney disease (CKD) is a progressive disorder that is characterized by the gradual loss of kidney function, often leading to end-stage renal failure. Recent research has highlighted the role of gut dysbiosis and its metabolic byproducts in the pathogenesis of CKD, with a particular focus on short-chain fatty acids (SCFAs). SCFAs, including acetate, propionate, and butyrate, are primarily produced by the fermentation of dietary fibers by the gut microbiota and are known for their systemic anti-inflammatory and immunomodulatory properties. In CKD, gut dysbiosis results in a reduction in SCFA-producing bacteria and an increase in uremic toxin-producing microorganisms, contributing to systemic inflammation, oxidative stress, and renal fibrosis. The depletion of SCFAs has been shown to exacerbate glomerular injury, whereas their presence supports integrity of the glomerular barrier and confers protection against damage. These protective effects are mediated by several mechanisms, including the modulation of immune responses, preservation of epithelial barrier function, and activation of specific receptors, such as G protein-coupled receptor 41 (GPR41), GPR43, and GPR109A. The present review provides a comprehensive overview of current understanding of SCFA-mediated pathways in glomerular protection during CKD progression. It highlights the therapeutic potential of targeting the gut–kidney axis to mitigate CKD progression by examining the complex interplay between gut microbiota and disease development, with a particular focus on strategies to protect the glomerular structure and function.

## 1. Introduction

Chronic kidney disease (CKD), characterized by a progressive and irreversible decline in kidney function, represents a major and growing global health burden. It is currently estimated that nearly 700 million people worldwide are affected by CKD [1]. According to the World Health Organization, kidney diseases have moved from the 19th to the 9th leading cause of death globally, with associated mortality nearly doubling between 2000 and 2021 [2]. Importantly, both prevalence and mortality continue to show upward trends, with projections suggesting further increases in the coming decade. The major risk factors for CKD include diabetes, hypertension, cardiovascular disease, and population aging, while environmental exposures, occupational risks, and dietary patterns are also emerging contributors. Moreover, there are substantial regional disparities: prevalence and mortality are highest in Asia, Africa, and Latin America, reflecting the combined impact of rapid population growth, limited healthcare access, and high burdens of diabetes and hypertension. These patterns underscore that CKD is not only a global health crisis but also one shaped by profound regional and socioeconomic inequalities [1,3].

CKD progresses slowly, often remaining asymptomatic until its advanced stages. Despite its widespread implications, the early detection and management of this condition remain challenging because of the lack of predictive, noninvasive biomarkers for early diagnosis. Additionally, the lack of treatment at this stage allows CKD to progress to end-stage renal disease (ESRD), where kidney function is so severely compromised that dialysis or kidney transplantation becomes necessary to sustain life. Glomerular diseases are a key driver of CKD through the central role of glomeruli in the filtration process, and their damage is closely linked to the onset of proteinuria, renal inflammation, and ultimately the deterioration of renal function. Understanding the mechanisms that contribute to glomerular injury is critical for developing strategies to prevent or slow the progression of CKD. Given the complex and overlapping pathological processes involved in glomerular diseases, identifying therapies that can target these diverse mechanisms, particularly in the early stages, remains a high clinical priority. Among the promising candidates are gut-derived metabolites, particularly short-chain fatty acids (SCFAs), for which emerging evidence suggests multifaceted roles in modulating kidney health and disease [4,5,6]. SCFAs, such as acetate, propionate, and butyrate, are produced by the fermentation of dietary fibers by the gut microbiota. Traditionally known for supporting gut health and metabolism, SCFAs also exert systemic anti-inflammatory and immunomodulatory effects [7], with pleiotropic actions on various organs [8,9], including the kidneys [10]. In the context of kidney disease, they are emerging as potential modulators of key pathological processes such as glomerular injury [11], fibrosis [12], and inflammation [13].

This review aims to synthesize current knowledge on the protective roles of SCFAs in CKD, with a particular focus on their mechanistic contributions to glomerular health, maintenance of barrier integrity, regulation of metabolism and oxidative stress, modulation of inflammation, and attenuation of fibrosis. By aligning the discussion with the therapeutic potential of SCFAs, a translational perspective is provided on how modulation of gut-derived metabolites may contribute to future CKD management strategies.

## 2. Methods

To ensure a focused and high-quality review, we performed a targeted search in PubMed using search terms including ‘short-chain fatty acids’, ‘SCFA’, ‘glomeruli’, ‘podocyte’, and ‘chronic kidney disease’ with no restrictions on publication date. Studies were included if they addressed mechanistic roles of SCFAs in the glomerular compartment, notably podocyte function, in the context of CKD. Titles and abstracts were screened first, with relevant studies undergoing full-text review to confirm eligibility.

## 3. Gut–Kidney Axis

A growing body of evidence confirms the commensal relationship between the gut microbiota and kidney [14]. The gastrointestinal tract, especially the large intestine, harbors a diverse and complex microbial community of trillions of microorganisms that are symbiotically connected with the host. In healthy adults, the gut microbiota is dominated by the phyla *Firmicutes* and *Bacteroidetes*, which together comprise approximately 90% of all species, followed by *Proteobacteria* and *Actinobacteria* [15,16]. Under normal physiological conditions, composition of the gut microbiota remains stable and in dynamic balance with the host. This microbiota is involved in various physiological processes, including energy metabolism, metabolic signaling [17], and the regulation of intestinal barrier integrity [18]. Additionally, microbial metabolites play a key role in regulating local and systemic immunity, helping to maintain immune homeostasis [19]. However, external factors, such as changes in diet, the use of antibiotics and medications, chronic stress, and pathogenic infections, can disrupt the intestinal microenvironment. Such disturbances can lead to dysbiosis, a condition that is characterized by an imbalance in the microbial community. Dysbiosis has been associated with the development of inflammatory diseases, typically marked by an increase in uremic toxin-producing microorganisms and a reduction in beneficial SCFA-producing bacteria [20,21]. SCFAs, such as acetate, propionate, and butyrate, are produced through the bacterial fermentation of undigested carbohydrates, primarily by *Bacteroidetes* and *Firmicutes*. These SCFAs are primary nutritional sources for colonic epithelial cells and help maintain integrity of the epithelial barrier by regulating tight-junction proteins. An intact protective epithelial layer keeps the microbiota separate from the host’s immune cells and prevents the translocation of toxic microbial waste products from the intestinal lumen to systemic circulation. When gut homeostasis is disrupted, the production of uremic toxins that derive from bacterial metabolism, such as indoxyl sulfate, p-cresyl sulfate, trimethylamine-*N*-oxide, and indole-3 acetic acid, increases. These toxins increase intestinal permeability and infiltrate the bloodstream, reaching extraintestinal sites, including the kidney. The renal accumulation of uremic toxins triggers the release of proinflammatory cytokines, accelerating the progression of kidney disease. This pathogenic interplay between the gut microbiota and kidney diseases is referred to as the gut–kidney axis [22,23] (Figure 1).

## 4. SCFAs: Overview

SCFAs are straight-chain saturated fatty acids with fewer than six carbon atoms. The most abundant SCFAs in the human intestinal tract are acetic acid (C2), propionic acid (C3), and butyric acid (C4), which are present in an approximate molar ratio of 60:20:20, respectively, in the colon and blood [24]. These SCFAs are produced by the anaerobic fermentation of undigested carbohydrates that derive from dietary fibers, including nonstarch polysaccharides, oligosaccharides, and resistant starch [25]. As microbial fermentation products with bioactive properties, SCFAs are classified as postbiotics, mediating diverse metabolic and immunomodulatory effects [26]. SCFAs provide 60–70% of the energy that is required by colonocytes, and butyrate is a significant energy source for these cells [27]. In addition to existing in the colon, SCFAs can be absorbed into the portal vein and enter the peripheral circulation, where they are utilized by various tissues. A small fraction of SCFAs can cross the epithelial barrier in an un-ionized form directly. Approximately 80–90% exist in an ionized form and require specialized transporters for absorption. This active transport is mediated by two types of transporters: monocarboxylate transporters and sodium-coupled monocarboxylate transporters, which are highly expressed in colonocytes and along the overall gastrointestinal tract [28]. In mammals, circulating SCFAs are considered important substrates and regulatory factors for energy metabolism [17]. Acetate, propionate, and butyrate act as precursors for lipid and glucose metabolism, mainly through the citric acid cycle or β-oxidation in mitochondria after conversion to acetyl-coenzyme A. SCFAs have been shown to improve hepatic and whole-body glucose homeostasis by regulating insulin-dependent hormones or promoting intestinal gluconeogenesis, which leads to satiety and lower hepatic glucose production [29]. Additionally, SCFAs reduce peripheral lipid accumulation by stimulating fatty acid oxidation and decreasing de novo lipogenesis and lipolysis [17]. SCFAs have been shown to reduce plasma concentrations of cholesterol [30]. In addition to serving as an energy source, SCFAs regulate metabolism through two major signaling mechanisms [31]. First, SCFAs, particularly butyrate, inhibit histone deacetylase (HDAC) activity, promoting histone acetylation and influencing the expression of genes that are related to cell proliferation, differentiation, and the inflammatory response. Second, SCFAs act as signaling molecules through G protein-coupled receptors (GPCRs), including G protein-coupled receptor 109A (GPR109A), GPR41, and GPR43. Upon activation by their specific ligands, these GPCRs interact with different G proteins to activate various intracellular signaling pathways, modulating such effectors as adenylyl cyclase and phospholipase C and leading to changes in second messenger levels, including cyclic adenosine monophosphate (cAMP) and intracellular calcium. Butyrate primarily activates GPR109A [32] and has high affinity for GPR41, whereas propionate acts as a potent agonist of both GPR41 and GPR43 [33]. Acetate is more selective for GPR43. An increasing body of evidence suggests that SCFAs, acting through these receptors, play a pivotal role in modulating physiological functions and alleviating inflammatory responses within renal tissue [10,34].

## 5. Structure of the Renal Glomerulus and Its Alterations in CKD

The mammalian kidney plays a crucial role in filtering blood and concentrating metabolic waste into the urine, thereby maintaining fluid homeostasis and regulating osmotic pressure. This filtration process primarily occurs in complex structures, known as renal glomeruli. The glomerular capillary barrier is a highly specialized structure that restricts the passage of large molecules and serum albumin into Bowman’s space while remaining permeable to small molecules and water. The mature glomerulus consists mainly of podocytes, glomerular endothelial cells (GEnCs), and mesangial cells. These interdependent cells work together to sustain integrity and function of the glomerulus [35]. Podocytes are terminally differentiated epithelial cells with intricate projections, called foot processes (FPs), that wrap around glomerular capillaries, forming filtration slits that are covered by slit-diaphragm (SD) proteins [36]. The SD structure allows communication between neighboring podocytes and acts as a size selectivity barrier for blood filtration. Glomerular endothelial cells are highly specialized cells that are covered by a negatively charged glycocalyx layer that can selectively restrict the passage of negatively charged molecules, such as albumin, and limit the adhesion of leukocytes and platelets. Together with podocyte SDs and the glomerular basement membrane (GBM), these cells create an interconnected glomerular filtration barrier [36,37]. This structure is a crucial component of the glomerular capillary wall and plays a vital role in regulating size selectivity of the glomerular filter. Mesangial cells are located between glomerular capillary loops and, together with the extracellular matrix (ECM), form the mesangium, which provides structural support to the glomerular vasculature and regulates capillary blood flow. CKD generally arises from disruption of the kidney filtration barrier, characterized by persistent inflammation, fibrosis, and the progressive loss of renal function, ultimately leading to protein leakage into urine and ESRD. Proteinuria is preceded by various changes, including glomerular expansion (which causes endothelial dysfunction and hemodynamic changes), the loss of glomerular basement membrane electric charge and its thickening, a decrease in the number of podocytes, FP effacement, and mesangial distension that leads to the loss of glomerular filter capacity [37]. Proteinuric kidney diseases arise from a wide range of disorders, but the leading cause of ESRD and progressive renal failure is diabetic kidney disease (DKD), one of the most severe microvascular complications of diabetes mellitus (DM) [38]. Among other causes are hypertension, glomerulonephritis, rheumatologic immune system diseases (e.g., systemic lupus erythematosus), infections, and environmental exposure [39].

## 6. Multi-Targeted Mechanisms of SCFA-Mediated Renoprotection

### 6.1. Relationship Between Endogenous SCFAs and Renal Function in CKD

An accumulating body of evidence supports the existence of a bidirectional relationship between the intestinal microbiome and kidney function. Dysbiosis is considered a central player during the manifestation of early pathophysiological changes in CKD [40]. Specifically, CKD development is linked to alterations of the gut microbiome, including the expansion of bacterial families that possess urease and uricase enzymes and a reduction in SCFA-producing families [20]. SCFAs that derive from gut microbes exert anti-inflammatory effects. Low levels of these beneficial microbes and their metabolites trigger a cascade of inflammatory, apoptotic, and fibrotic pathways, contributing to the progression of renal pathology and glomerulopathies. A growing body of research highlights a strong correlation between low SCFA levels and worse kidney function. For example, Wang et al. reported that CKD patients had low serum SCFA levels [41]. Similarly, Corte-Iglesias et al. reported that propionic and butyric acid levels gradually decreased in CKD patients as the disease progressed, closely correlating with clinical parameters of renal function [42]. In individuals with immunoglobulin A nephropathy, reductions in SCFAs levels were found, which negatively correlated with indicators of kidney injury [43]. Specifically, low butyric acid levels were associated with high levels of uric acid and blood urea nitrogen (BUN). These trends also extend to DKD patients. Metabolomics analyses revealed a significant decrease in serum butyrate in individuals with DN compared with healthy subjects [44]. Cai et al. [45] reported that patients who underwent dialysis had low serum levels of acetate, propionate, butyrate, and total SCFAs compared with healthy individuals and type 2 DM (T2DM) patients without renal injury. These metabolites positively correlated with the estimated glomerular filtration rate (eGFR) and negatively correlated with the urinary albumin-to-creatinine ratio (UACR). Additionally, fecal levels of acetate, propionate, butyrate, and total SCFAs were significantly lower in DKD patients [24], showing positive correlations with the eGFR and an inverse relationship with BUN levels. Animal studies reinforced these findings. For example, streptozotocin-induced diabetic rats exhibited low fecal and serum butyrate levels [45]. These compelling observations suggest that gut dysbiosis and a decrease in SCFA-producing bacteria may lower serum SCFAs levels, potentially accelerating the progression of kidney diseases. As a result, SCFAs, particularly butyrate, are emerging as promising therapeutic targets for CKD treatment.

### 6.2. Effect of Supplemental SCFAs on Renal Function in CKD

Several studies suggest that both naturally occurring butyrate and supplemental sodium butyrate may exert protective effects against renal injury. SCFA supplementation has been shown to reduce proteinuria and alleviate kidney damage in experimental models of autoimmune glomerulonephritis [46]. Sodium butyrate treatment increased serum butyrate levels and improved renal damage in diabetic mice [45]. In diabetic db/db mice that were treated with butyrate, researchers found significant reductions in serum creatinine, BUN, and the UACR [44]. In diabetic rats, sodium butyrate reduced both glomerular area and the kidney-to-body weight ratio compared with untreated diabetic rats, demonstrating a clear protective effect on kidney structure. Moreover, the early administration of propionate and butyrate in a preclinical mouse model of nephropathy that was induced by folic acid injury resulted in the long-term preservation of renal function and prevented the development of CKD [42]. Propionate has also demonstrated antihypertensive properties, in which propionate supplementation prevented hypertension in offspring that was induced by maternal CKD [47]. Collectively, these findings suggest that SCFAs could be promising therapeutic agents for the prevention and treatment of CKD, based on their ability to reduce proteinuria and prevent renal failure. Their potential to modulate inflammation and fibrosis offers new hope in combating renal pathology and improving patient outcomes (Figure 2).

### 6.3. Effect of Supplemental SCFAs on Levels of Inflammatory Cytokines in CKD

Inflammation is a critical pathophysiological process that is involved in the pathogenesis of glomerular diseases. Immune system disturbances and inflammatory responses contribute to the onset and progression of CKD and its associated complications [48]. Uncontrolled inflammation often results from leukocyte infiltration into damaged tissues, leading to the release of proinflammatory cytokines, such as interleukin-1β (IL-1β), IL-6, and tumor necrosis factor-α (TNF-α). Recent research highlights the critical role of SCFAs in regulating immune responses and inflammation [18]. In patients who were undergoing maintenance hemodialysis, sodium propionate supplementation significantly reduced key serum biomarkers of inflammation, including C-reactive protein, IL-2, IL-6, IL-10, IL-17a, TNF-α, interferon-γ (IFN-γ), and transforming growth factor-β (TGF-β) [6]. These clinical outcomes are consistent with findings from animal models, in which SCFAs exerted strong anti-inflammatory effects on renal tissue, highlighting their therapeutic potential in managing kidney disease. For example, sodium butyrate has been shown to alleviate TNF-α and IL-6 expression while suppressing the Toll-like receptor signaling pathway in renal tissue in lipopolysaccharide (LPS)-treated rats [49]. In a mouse model of DKD, sodium butyrate reduced serum IL-6 levels [5] and kidney tissue levels of TNF-α, vascular cell adhesion molecule-1, and intercellular cell adhesion molecule-1 [50]. Similarly, sodium acetate decreased the gene expression of IL-6 and IFN-γ in kidney tissue [10]. Huang et al. further demonstrated reductions in IL-1β levels in the kidneys in T2DM mice that were treated with SCFAs, underscoring SCFAs’ protective role against inflammation and renal damage [13].

In cellular models of diabetes, SCFAs reduced IL-6 expression in glomerular mesangial cells (GMCs) [5], tubular epithelial cells, and podocytes [10] under hyperglycemic conditions. Specifically, butyrate and propionate decreased TNF-α levels in podocytes [10]. Notably, the protective effect of butyrate against podocyte injury involved GPR109a, supporting the hypothesis that the butyrate–GPR109a axis plays a pivotal role in modulating renal inflammatory responses. The nuclear factor-κB (NF-κB) signaling pathway is crucial for the release of many chemokines and cytokines that are related to inflammation. SCFAs, especially butyrate, have been shown to extinguish NF-κB signaling in the kidneys in diabetic mice and GMCs that were cultured in high glucose [13]. These effects were mediated by GPR43 activation and the interaction between β-arrestin-2 and I-κBα, suggesting a complex yet powerful anti-inflammatory mechanism of SCFAs in protecting renal function [13].

### 6.4. Effect of Supplemental SCFAs on Chemotaxis Inflammatory Cells

SCFAs are well known for their anti-inflammatory properties by modulating immune cell adhesion, chemotaxis, and cytokine release, ultimately inhibiting leukocyte migration to inflammatory sites [51,52]. In a mouse model of ADR-induced nephropathy, butyrate significantly reduced the infiltration of CD68-positive cells and lowered the expression of chemokines like C-C motif ligand 2 (CCL2) and CCL3 [11]. These cytokines, also known as monocytic chemotactic protein 1 (MCP-1) and macrophage inflammatory protein-1α, respectively, are members of the CC-chemokine family and chemotactic factors that recruit inflammatory cells for the immune response and maintain an effector inflammatory reaction [53,54]. In diabetic mice, SCFA supplementation also reduced MCP-1 levels, with butyrate altering serum levels [5] and acetate decreasing its mRNA expression in the kidneys [10]. SCFA treatment also lowered MCP-1 levels in the kidneys in T2DM mice and high glucose-exposed GMCs [13]. Additionally, butyrate and acetate suppressed the synergistic effect of high glucose and LPS on MCP-1 release by GMCs [55]. Interestingly, Li et al. [56] found a compelling link between renal injury in T2DM and low SCFA production, which impaired the anti-inflammatory roles of endogenous SCFAs, particularly through the C5a-C5aR axis. The interaction between complement protein C5a and its receptor is a crucial component of the innate immune system, driving leukocyte infiltration into damaged tissue and triggering the release of proinflammatory cytokines, such as IL-1β, IL-6, and TNF-α. This study highlighted greater activation of the complement C5a pathway in the kidneys in diabetic mice and DKD patients. Inhibiting the C5a pathway restored gut microbiota diversity and increased fecal levels of acetate, butyrate, and propionate in db/db mice. Furthermore, SCFA treatment suppressed the inflammatory response in GEnCs similarly to inhibiting the C5a pathway [56]. These findings suggest that SCFAs could play a crucial protective role against kidney injury in T2DM by preventing the excessive activation of complement C5 and reducing the chemotaxis of inflammatory cells, thereby mitigating inflammation and tissue damage.

### 6.5. Role of Dietary Fiber in the Inflammatory Response in CKD

In addition to beneficial effects of exogenous butyrate, growing interest has been seen in diet as a major external factor that influences gut microbiota composition, offering a potential therapeutic approach to restore intestinal symbiosis [57,58]. Dietary fiber, which is known to decrease inflammation and mortality in CKD [59,60], is believed to exert its effects through SCFAs. In diabetic mice, a diet that contained enriched amounts of fiber attenuated the infiltration of CD68-positive cells [10] and lowered CCL2 expression in renal tissue. Mice that were fed a high-fiber diet were also partially protected from the development of acute kidney injury (AKI) and its progression to CKD, exhibiting a decrease in the expression of proinflammatory cytokines (e.g., IL-6, TNF, and IL-1), chemokines (e.g., CCL2, chemokine ligand 2 [CXCL2], and CXCL10), inflammatory mediators (e.g., inducible nitric oxide synthase [iNOS]), and inflammasome components (e.g., NACHT, LRR, PYD domains-containing protein 3 [NLRP3], and IL-1 [61]). High-fiber intake significantly decreased the accumulation of renal macrophages and T-cells, effects that were similarly observed with SCFA supplementation [61]. Supporting these findings, a study by Tayebi-Khosroshahi et al. [62] reported lower serum levels of inflammatory markers (i.e., TNF-α and IL-6) in hemodialysis patients who received a high-amylose diet. In summary, SCFAs that derive from the gut microbial fermentation of dietary fiber have strong anti-inflammatory and immunomodulatory properties by reducing the recruitment of inflammatory cells and suppressing the production of such cytokines as IL-6, IL-1β, and TNF-α. These promising findings underscore the therapeutic potential of dietary interventions, particularly fiber, in modifying renal inflammatory responses.

### 6.6. Oxidative Stress in CKD: Overview

Oxidative stress is a condition in which the production of reactive oxygen species (ROS) and reactive nitrogen species (RNS) exceeds the capacity of the antioxidant system, resulting in tissue damage and dysfunction. In the kidneys, ROS are mainly produced by cytosolic enzymes, such as NADPH oxidase (NOX), and the mitochondrial respiratory chain. Excessive ROS production can be neutralized by scavenger systems, including antioxidant factors such as superoxide dismutase (SOD), glutathione peroxidase (GPX), catalase (CAT), and nitric oxide (NO). Impairments in antioxidant capacity result from defective activation of the transcription factor nuclear factor erythroid 2-related factor 2 (Nrf2), which is the master regulator of nearly 250 genes that encode antioxidant and cytoprotective enzymes and proteins [63,64]. Under physiological conditions, free radicals play a role in signal transduction; however, under unfavorable conditions, unconstrained ROS and RNS production results in oxidation and damage to biological molecules, such as lipids, proteins, and DNA [65,66]. The end products of these reactions serve as biomarkers for measuring oxidative stress in various tissues and biological samples. For example, malondialdehyde (MDA) is a byproduct of lipid peroxidation, and 3-nitrotyrosine (3-NT) is formed by the nitration of tyrosine residues in proteins. The kidney is a highly metabolic organ, rich in oxidation reactions in mitochondria, making it vulnerable to damage that is caused by oxidant radicals that accelerate kidney disease progression [65].

### 6.7. Oxidative Stress: Role of SCFAs in CKD

Gut microbiota disruption leads to excess uremic toxins and reduced SCFAs, driving uremia, inflammation, oxidative stress, and CKD progression [67,68]. Numerous studies have shown that sodium butyrate can improve intestinal barrier function and inhibit kidney inflammation by restoring oxidative balance [4,34,69,70]. In GMCs that were exposed to high glucose and LPS, SCFA treatment reduced redox imbalance, by lowering intracellular ROS and MDA production while increasing SOD activity [55]. SCFAs also prevented activation of the NF-κB cascade in these cells that were exposed to high glucose [13]. Effects of butyrate were significantly inhibited by GPR43 siRNA and enhanced by GPR43 overexpression, highlighting the critical role of GPR43 in mediating these protective effects.

Similar antioxidant effects of butyrate have been demonstrated in animal models. In diabetic mice, sodium butyrate reduced kidney oxidative stress by lowering the expression of NOX4, an enzyme that is linked to ROS production [69]. Butyrate supplementation also protected against renal oxidative damage by lowering various markers, including MDA, iNOS, and 3-NT [50]. The antioxidant response to butyrate was found to depend on NRF2 and its downstream antioxidant genes, including heme oxygenase 1 (HO-1) and NAD(P)H dehydrogenase quinone 1 (Nqo1). Thus, SCFAs, especially butyrate, can positively influence diabetic kidney injury by inhibiting oxidative stress and NF-κB signaling via GPR43 and may be potential therapeutic agents for the prevention and treatment of DN.

### 6.8. Role of Dietary Fiber in Oxidative Stress in CKD

The role of dietary fiber in reducing oxidative stress and inflammation has gained significant attention as a promising therapeutic avenue for kidney disease. Vaziri et al. [71] reported a protective role for dietary fiber in redox signaling in the diabetic kidney through the NRF pathway. Diabetic mice that were fed a high-fiber diet exhibited an increase in the expression of antioxidant defense systems, including NRF and its downstream targets, such as GPX and HO-1. Additionally, the fiber-rich diet reduced markers of oxidative damage, such as 3-NT, and enzymes that generate free radicals, such as NAD(P)H oxidase subunits (e.g., NOX-4 and gp91phox).

Clinical trials confirmed the benefits of high-fiber diets in reducing oxidative stress in ESRD patients. Tayebi-Khosroshahi et al. [62] found that hemodialysis patients who consumed a diet that was enriched with amylose (HAM-RS2) exhibited a reduction in serum MDA levels. Similarly, dietary fiber supplementation in hemodialysis patients lowered serum MDA levels and improved total antioxidant capacity [6]. These findings highlight beneficial effects of SCFAs and dietary fiber supplementation in reducing oxidative damage in kidney tissue in both animal models and clinical studies. This underscores the potential of a high-fiber diet as an essential approach for preventing CKD progression and supporting kidney health.

### 6.9. Role SCFAs in Mitochondrial Oxidative Stress

Oxidative stress and excess ROS/RNS impair mitochondrial function by disrupting the electron transport chain and reducing ATP synthesis. In CKD, dysfunctional mitochondria become key sources of ROS, amplifying injury and inflammation [72]. Nicese et al. [73] found that sodium butyrate enhanced mitochondrial biogenesis and prevented mitochondrial fragmentation, a key factor that underlies mitochondrial dysfunction in kidney injury. The formation of new mitochondria is crucial for maintaining energy balance and proper cellular metabolism, particularly under stress conditions. Butyrate’s ability to support cellular energy helps preserve cellular function during oxidative stress. Additionally, in diabetic mice, butyrate restored mitochondrial biogenesis via the AMP-activated protein kinase ((AMPK)/peroxisome proliferator-activated receptor γ coactivator 1-α (PGC-1α) signaling pathway) [74]. Butyrate’s positive effects on mitochondria were accompanied by a decrease in ROS levels, an increase in ATP content, and an improvement in the NADP+/NADPH ratio. Overall, these findings suggest that butyrate is a potential therapeutic agent for mitigating mitochondrial damage and preserving kidney function.

### 6.10. Role of SCFAs in Glucose Metabolism and Strengthening Barrier Integrity in CKD

A growing body of evidence suggests that improvements in inflammation and oxidative stress by SCFA treatment are closely linked to greater glucose metabolism. In hemodialysis patients, propionic acid supplementation was shown to reduce fasting insulin levels and improve the HOMA index (homeostasis model assessment), indicating better insulin sensitivity [6]. Similarly, in diabetic mice, butyrate supplementation improved both insulin sensitivity and glucose tolerance [74]. Gonzalez et al. [4] proposed that butyrate’s protective effects on renal function arise from its ability to improve glucose metabolism and enhance intestinal integrity. Their research in CKD rats revealed that butyrate treatment strengthens the intestinal barrier while also stimulating the secretion of glucagon-like peptide-1 (GLP-1) and increasing the phosphorylation of AMPK in the colon. GLP-1, a key metabolic hormone, activates AMPK, promoting glucose homeostasis, enhancing insulin sensitivity, and regulating lipid metabolism. AMPK also plays a vital role in maintaining barrier integrity by regulating tight-junction proteins, responding to energy depletion and oxidative stress, and stabilizing cytoskeletal structure in both intestinal [75] and glomerular [76,77] cells. Therefore, butyrate could be a promising therapeutic agent by effectively targeting the AMPK pathway, enhancing glucose metabolism, and preserving intestinal barrier integrity in CKD patients. Additionally, Zhou et al. [34] found that the AMPK signaling pathway and glucose metabolism coupled with GPR43 activation in the colon. This dependence restores intestinal permeability and prevents systemic inflammation, leading to a reduction in lipid deposition and the suppression of inflammation and oxidative stress in the kidney [34]. Other research by Nicese et al. [73] revealed the ability of butyrate to enhance the monolayer resistance of GEnCs by increasing the expression of claudin-5 and VE-cadherin, major components of tight and adherens junctions in vascular endothelial cells. Similarly, a high-fiber diet restored the integrity of gut epithelial tight junctions in CKD rats [71], demonstrating the efficacy of a high-fiber diet in minimizing this abnormality. Therefore, effects of butyrate on glucose regulation and barrier integrity could be essential targets in managing hyperglycemia and improving barrier function in metabolic and kidney diseases. Strengthening the intestinal barrier can effectively protect against the translocation of microbial uremic toxins, which are known to contribute to systemic inflammation and worsen kidney function. In parallel, maintaining integrity of the glomerular barrier in the kidneys prevents protein leakage into urine. This dual approach helps prevent further damage to kidney function by limiting toxin exposure and reducing proteinuria.

Adverse effects of GPR43 on glucose metabolism and glomerular function were demonstrated in diabetic mice and podocytes exposed to high glucose conditions [78]. The authors observed that GPR43 expression was significantly upregulated under diabetic conditions, which corresponded with impaired insulin signaling and pathological alterations in kidney morphology. Notably, deletion of GPR43 alleviated albuminuria, prevented mesangial expansion, preserved podocyte number, and restored glucose metabolism. Increased expression of GPR43 may be driven by alterations in the gut microbiota composition in diabetes that resulted in overproduction of acetate, a key SCFA that activates GPR43. These findings highlight the critical role of gut microbiota dysbiosis in modulating host metabolic signaling pathways and contributing to the pathogenesis of diabetic complications [78]. Although the current literature predominantly highlights the beneficial effects of SCFAs on the host and kidney function, the potential pathogenic link between gut microbiota dysbiosis and altered GPR signaling suggests the need for a more personalized approach to diagnosis and therapy.

### 6.11. Kidney Fibrosis: Overview

Kidney fibrosis, the final manifestation of CKD, is characterized by the loss of renal cells and their replacement by the ECM. This fibrogenic process involves various factors, including inflammatory agents, cytokines, vasoactive agents, and enzymes that are responsible for ECM assembly, anchoring, and degradation. Inflammatory stimuli provoke the activation of mesangial cells and fibroblasts, leading to the excessive production of ECM components, such as fibronectin, type I and IV collagens, laminin, proteoglycans, and α-smooth muscle actin (α-SMA). The crucial cytokine in this process is TGF-β, which is secreted by macrophages in response to tissue injury. TGF-β initiates the release of profibrotic mediators, such as plasminogen activator inhibitor type-1 (PAI-1), α-SMA, and connective tissue growth factor (CTGF), driving fibrosis, which affects all kidney compartments with glomerular fibrosis, referred to as glomerulosclerosis [79,80,81].

### 6.12. Kidney Fibrosis: Role of SCFAs in CKD

Increasing evidence has emphasized the significant role of SCFAs as emerging therapeutic targets for renal inflammatory diseases and kidney fibrosis [10,12,42]. For example, butyrate has been shown to reduce glomerulosclerosis in mice with Adriamycin (ADR)-induced nephropathy by decreasing levels of proteoglycans, collagen deposition, and the expression of fibrotic markers, including TGF-β and collagen 4α1 (COL4α1), compared with untreated animals [11]. Similarly, in diabetic mice, the administration of sodium butyrate prevented glomerular enlargement and mesangial matrix expansion and reduced the elevation of protein levels of key fibrotic mediators, including PAI-1 and CTGF [50]. Li et al. reported therapeutic effects of acetate and butyrate in improving glomerular histomorphology and reducing fibrosis in diabetic mice [82]. Furthermore, GPR43 and GPR109a were identified as critical for SCFA-mediated protection, in which mice that lacked these receptors did not exhibit the same level of renoprotection [82]. This underscores the therapeutic potential of SCFAs in mitigating kidney fibrosis through their actions on these receptors.

### 6.13. Role of SCFAs in Podocyte Fibrosis

Podocytes play a crucial role in the development of glomerular sclerosis through their limited ability to proliferate. Extensive podocyte injury and loss from the GBM trigger a compensatory response, including the proliferation of renal parietal cells and deposition of ECM components to replace lost podocytes [83,84]. Felizardo et al. [11] conducted an ultrastructure analysis of glomeruli and found that treatment with butyrate in mice with ADR-induced nephropathy preserved the podocyte layer that surrounds the GBM, maintained the SD, and reduced FP effacement. The protective effect of butyrate in this podocyte injury model was further confirmed by the restoration of mRNA levels of nephrin and podocin, two key proteins of the structure of SDs. Consistent with these findings, the administration of a high-butyrate-releasing diet to the same mice reduced the loss of Wilm’s tumor 1 (WT-1) stained podocytes [11]. These findings indicate that butyrate’s protective effect on podocytes operates through GPR109a. Furthermore, butyrate restored mRNA expression levels of transient receptor potential channel 5 (TRPC5), TRPC6, cell division control protein 42 (Cdc42), and RhoA in podocytes. The modulation of RhoA, Cdc42, and Rac1 activity is known to influence cytoskeletal remodeling, which is crucial for preventing FP effacement and maintaining the filtration barrier [85,86]. These findings suggest a possible link between GPR109a signaling in podocytes and the regulation of calcium-dependent cytoskeletal remodeling, controlled by small guanosine triphosphatases, which govern the formation of lamellipodia and filopodia [11]. This is particularly important in the context of FP effacement and the pathogenesis of proteinuric glomerular diseases. Li et al. [82] also reported the protective effect of SCFAs on podocytes through GPR43. The in vitro exposure of podocytes to high glucose stimulated profibrotic gene expression, whereas treatment with acetate or butyrate suppressed these responses. However, both SCFAs were less effective in podocytes that were genetically deficient in GPR43, highlighting the critical role of this receptor in mediating protection against fibrosis [82]. In a subsequent study, the same researchers confirmed that acetate, propionate, and butyrate reduced profibrotic factors, such as fibronectin and TGF-β, in podocytes under hyperglycemic conditions [10]. These findings emphasize the crucial role of SCFAs, particularly butyrate, in protecting podocytes from injury and fibrosis through GPR109a and GPR43, thereby preserving the filtration barrier and preventing the progression of glomerular diseases.

### 6.14. Role of Dietary Fiber in Preventing Kidney Fibrosis

Dietary fiber has been shown to directly reduce renal fibrosis, a major contributor to the progression of CKD [5,10]. A fiber-enriched diet that was fed to diabetic mice prevented glomerular hypertrophy and mesangial matrix and collagen deposition and reduced the expression of fibrosis-related genes, such as TGF-β and fibronectin [10]. Furthermore, a fiber-rich diet preserved podocyte numbers in glomeruli. Similarly, a diet that was high in resistant starch in a rodent model of CKD attenuated abnormalities that are associated with interstitial fibrosis and reduced the upregulation of TGF-β, PAI-1, and α-SMA, suggesting deactivation of the fibrotic pathway [71]. These results highlight the potential of dietary fiber to modulate the gut microbiota and enhance the production of SCFAs, offering a promising strategy for preventing fibrosis and preserving kidney function.

### 6.15. Epigenetic Modifications in the Context of CKD and SCFAs

Growing interest has been seen in the role of epigenetic changes, such as DNA methylation, histone modifications, and non-coding RNA regulation, in the development of fibrosis and progression of glomerular diseases. Among these epigenetic mechanisms, SCFAs, particularly butyrate, have been shown to exert protective effects on the kidneys by influencing gene expression and reducing fibrosis. Du et al. [87] reported that butyrate alleviated kidney fibrosis through the regulation of non-coding RNA, specifically by modulating the miR-7a-5p/P311/TGF-β1 pathway, in db/db mice. Additionally, butyrate was shown to counteract renal inflammation and fibrosis in DKD through histone modification, specifically by promoting histone lysine butyrylation (Kbu) [5]. These histone modifications, together with the inhibition of HDACs, represent promising therapeutic strategies to slow the progression of renal fibrosis. Histone deacetylases, which remove acetyl groups from acetylated histones, compress chromatin and repress gene transcription. In the context of renal fibrogenesis, HDACs play a critical role by regulating inflammatory and profibrotic gene expression. For example, the administration of sodium butyrate in juvenile diabetic rats decreased HDAC activity and reduced kidney fibrogenesis [88]. Similarly, in streptozotocin-induced diabetic mice, butyrate supplementation protected against renal inflammation and fibrosis by suppressing HDAC activity and regulating NRF2 expression at the transcriptional level [50]. In mice with folic acid nephropathy, a high-fiber diet or SCFA supplementation exerted anti-inflammatory effects and protected against acute kidney injury and subsequent fibrosis by inhibiting HDAC [61]. Over recent decades, HDAC inhibitors have emerged as a novel therapeutic class that protects against renal injury. Although most HDAC inhibitors are nonselective (i.e., they target multiple HDACs), there is growing evidence of the role of individual HDACs in the development of kidney diseases. For example, HDAC4 expression significantly increased in the kidneys in patients with focal segmental glomerulosclerosis [89], and the upregulation of HDAC2, HDAC4, and HDAC5 was observed in patients with DN and animal models of diabetes [89]. Interestingly, HDAC4 was identified as a central player in diabetes-related podocyte injury. SCFAs were shown to reduce the mRNA expression of HDAC3, HDAC4, HDAC5, HDAC7, and HDAC10 in kidney tissue in a mouse model of AKI [61]. Based on these findings, a reasonable direction for future research would be to investigate the specific involvement of SCFAs in the selective inhibition of individual HDACs, which could offer more targeted therapeutic strategies for the prevention of renal fibrosis and injury.

## 7. Clinical Evidence for SCFA Supplementation in CKD

Promising evidence from cellular and animal models is driving the translation of these effects to patients with CKD. Although the number and scope of human studies remain limited, early data provide initial insights into the therapeutic potential of SCFAs in this population, with representative trials summarized in Table 1. Propionate supplementation has been studied in hemodialysis cohorts [6,90]. In a 12-week intervention trial, oral sodium propionate (1 g/day) significantly reduced systemic inflammation and oxidative stress and improved insulin resistance [6]. In a 30-day study, propionate increased circulating Tregs and reduced C-reactive protein (CRP); however, these effects were reversible, with Treg frequencies returning to baseline two months after discontinuation [90]. Notably, this immunoregulatory effect was not observed in kidney transplant recipients, likely due to concomitant triple immunosuppression [91]. By contrast, a randomized, placebo-controlled trial of oral sodium butyrate (3.6 g/day for 12 weeks) in adults with type 1 diabetes and albuminuria did not significantly change inflammatory markers, albuminuria, or kidney function [92]. The therapeutic potential of these agents is being actively explored in clinical practice, with several ongoing registered trials summarized in Table 1. However, as many trials involve only short treatment durations, further studies are needed to assess their long-term clinical effects.

## 8. Future Directions and Limitations

A growing body of evidence supports the protective effects of SCFAs in CKD and highlights the therapeutic potential of microbiome-based interventions. This underscores the need for further research to overcome current limitations. Dietary modifications designed to restore microbial balance and enhance SCFA production, as well as direct SCFA supplementation, offer promising avenues for improving CKD management. However, a critical limitation often overlooked is the substantial interindividual variability in gut microbiota composition, which can strongly influence both treatment response and disease progression. This variability also extends to endogenous SCFA production, which depends on diet, medication use, and environmental factors. Consequently, one of the major challenges in this field is the heterogeneity of SCFA exposure. Even when individuals receive the same dietary intervention or SCFA supplementation, the resulting systemic levels may differ considerably due to differences in microbial capacity for fermentation, absorption efficiency, and host metabolism. Given these considerations, the implementation of a more personalized diagnostic and therapeutic strategy is both justified and necessary. Integrating fecal microbial analysis with targeted metabolomics profiling allows for the identification of individual dysbiosis patterns and the metabolic activity of gut microbes [96]. This combined strategy could guide tailored interventions to modulate the gut–kidney axis more effectively in CKD patients. In the context of highly variable gut microbiome profiles, microbiota transplantation therapies, particularly fecal microbiota transplantation (FMT), are emerging as potential interventions [97]. FMT seeks to restore microbial diversity, increase SCFA levels, and reduce systemic inflammation and oxidative stress, thereby contributing to renal protection and improved metabolic homeostasis in animal models of CKD [98]. In humans, FMT appears promising; however, its application in CKD remains largely experimental [97,99,100,101] and requires further clinical investigation through larger-scale studies.

Most of the current evidence supporting the protective role of SCFAs in CKD originates from basic research and preclinical studies, including both in vitro experiments and in vivo models. These studies have elucidated the precise mechanisms by which SCFAs contribute to improved kidney function, offering valuable insights into their potential as therapeutic agents. Nevertheless, important translational gaps remain between animal and human studies: doses and exposure profiles effective in cellular and rodent models may not be feasible or equally efficacious in patients, and experimental endpoints do not always correspond to clinical outcomes such as eGFR decline. For instance, in one clinical trial, butyrate supplementation failed to improve renal indicators such as GFR or UACR [92]. It is possible that the supplementation period required to elicit cellular-level effects differs from that needed to reverse established organ injury. This suggests that longer treatment durations may be necessary to achieve clinically meaningful renal benefits. Another challenge lies in receptor targeting. SCFAs interact promiscuously with several receptors (GPR43, GPR41, GPR109A) that vary in tissue distribution, signaling pathways, and species-specific expression. This complexity limits therapeutic development and underscores the need for standardized methodologies and receptor-specific pharmacologic tools. Finally, a major limitation is the small sample size of most clinical investigations, which restricts the generalizability of current findings to broader patient populations. Taken together, these issues underscore the need for well-designed, large-scale clinical trials to validate these preliminary findings. Despite promising developments, key challenges remain, including the absence of standardized dosing protocols, established treatment guidelines, and sufficient clinical data in human populations. To address these gaps, robust multicenter trials and comprehensive translational research are essential to define effective and safe dosing strategies, optimal delivery methods, and long-term safety and efficacy of microbiome-targeted interventions in CKD management.

## 9. Conclusions

In conclusion, SCFAs have emerged as key mediators in the gut–kidney axis, providing significant protective benefits against CKD progression, especially in the context of glomerular injury. SCFAs act through multiple mechanisms, including reducing inflammation, oxidative stress, and fibrosis, while also strengthening the glomerular and intestinal barriers. Additionally, their activation of key receptors, such as GPR41, GPR43, and GPR109A, underscores the therapeutic potential of targeting SCFA production and the gut microbiota in CKD management (Figure 3). More research is necessary to translate these findings to clinical practice, but the current evidence clearly demonstrates the promising role of SCFAs in preventing kidney damage and slowing CKD progression.

## Figures and Tables

**Figure 1 nutrients-17-02904-f001:**
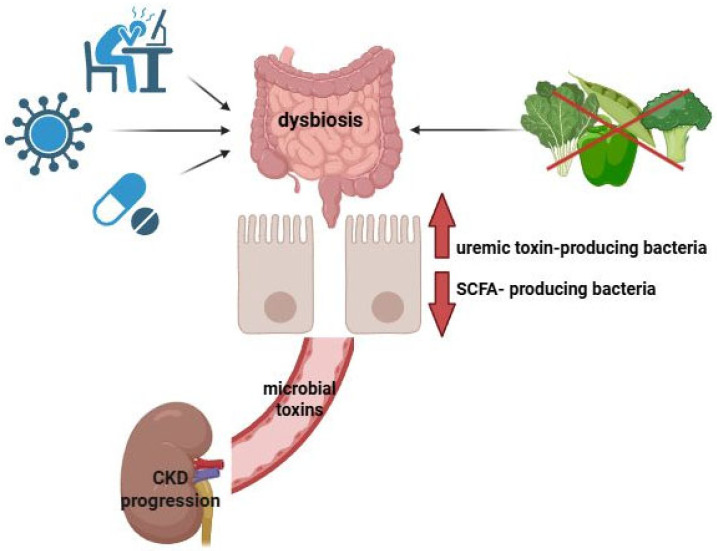
Gut–kidney axis. External factors, such as dietary changes, antibiotic and medication use, chronic stress, and pathogenic infections, can disrupt gut microbiota balance (dysbiosis), leading to an increase in uremic toxin-producing microorganisms and a reduction in beneficial SCFA-producing bacteria. The resulting bacterial toxins elevate intestinal permeability, allowing harmful substances to enter the bloodstream and reach extraintestinal sites, including the kidneys. This process triggers the release of proinflammatory cytokines, accelerating kidney disease progression. This pathogenic interplay is known as the gut–kidney axis. Created with BioRender.com.

**Figure 2 nutrients-17-02904-f002:**
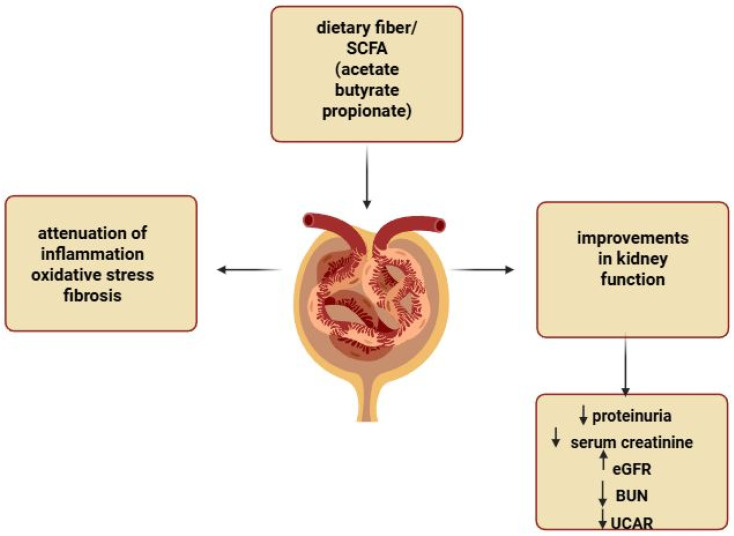
Overview of the effects of dietary fiber and SCFAs on kidney function. This figure demonstrates the protective role of dietary fiber and short-chain fatty acids (SCFAs), specifically acetate, butyrate, and propionate, in kidney health. SCFAs help to attenuate inflammation, reduce oxidative stress, and improve tissue fibrosis. These effects lead to improvements in kidney function, as shown by reductions in proteinuria, serum creatinine, BUN, and UCAR along with increased eGFR. “↓”—decreased; “↑”—increased. Created with BioRender.com.

**Figure 3 nutrients-17-02904-f003:**
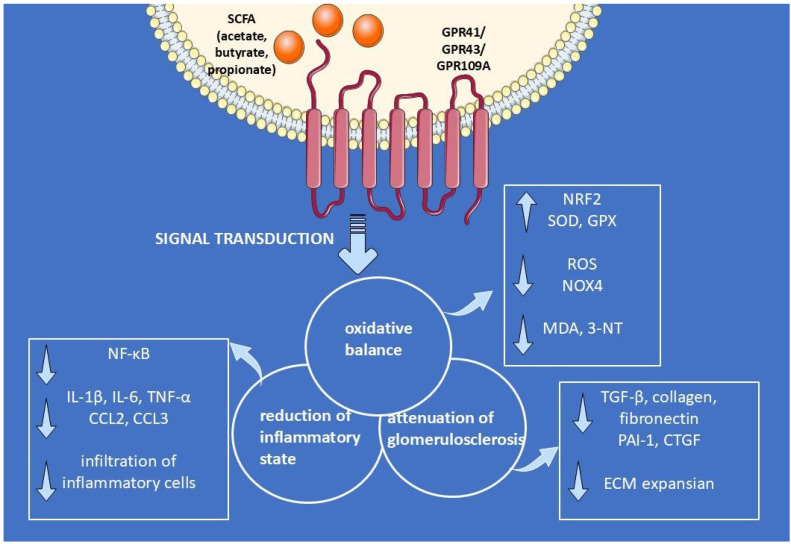
The role of SCFA-induced signaling pathways in kidney protection. This figure highlights the protective effects of short-chain fatty acids (SCFAs) through their interaction with GPR41, GPR43, and GPR109A receptors. Upon SCFA binding, these receptors trigger signal transduction pathways that enhance oxidative balance, reduce inflammation, and prevent the onset of glomerulosclerosis. SCFAs suppress the NF-κB cascade, leading to the downregulation of proinflammatory cytokines such as IL-1β, IL-6, TNF-α, CCL2, and CCL3. This cascade also decreases inflammatory cell infiltration into kidney tissues. In parallel, SCFAs enhance antioxidant defense systems, activating the NRF2 pathway and increasing expression of antioxidant enzymes like SOD and GPX. This helps reduce oxidative damage markers, including ROS, MDA, and 3-NT. SCFAs also inhibit the expansion of ECM by downregulating fibrosis-related factors such as TGF-β, collagen, fibronectin, PAI-1, and CTGF, thus limiting fibrotic tissue development and preserving kidney function. Created with smart.servier.com.

**Table 1 nutrients-17-02904-t001:** Clinical trials and intervention studies on SCFA administration in CKD patients.

Study (Year/ID)	Population/Demographics	Intervention and Route	Dose	Duration	Outcomes Measured	Key Findings/Conclusions	Type of Study
Completed/Published Interventions							
Marzocco et al., 2018 (PLAN) [6]	Adults on maintenance hemodialysis (*n* = 20)	Sodium propionate (oral capsules)	2 × 500 mg (1 g/day)	12 weeks (+4-week follow-up)	Markers of inflammation, oxidative stress and gut-derived toxins, biochemical parameters	Reduced inflammation/oxidative stress and uremic toxins; improved insulin resistance	Single-center non-randomized pilot study
Meyer et al., 2020 [90]	Hemodialysis (*n* = 10) and healthy volunteers (*n* = 7)	Sodium propionate (oral)	2 × 500 mg	30 days (+60-day follow-up)	Circulating Tregs, CRP, electrolytes, renal parameters	↑ Tregs, ↓ CRP	Prospective study
Anft et al., 2024 [91]	Hemodialysis (*n* = 10) and kidney transplant recipients (*n* = 16)	Propionic acid (oral)	2 × 500 mg	30 days (+60-day follow-up)	Tregs, immune cells analysis	Treg expansion in HD; absent in transplant recipients on triple immunosuppression	Prospective study
Tougaard et al., 2022 (RCT) [92]	Adults with T1D and albuminuria (*n* = 53; early DKD)	Sodium butyrate (oral) vs. placebo	3.6 g/day	12 weeks	Fecal calprotectin and SCFAs, CRP, UACR, eGFR, HbA1c	No significant differences vs. placebo for inflammatory or kidney endpoints	Randomized, double-blind, placebo-controlled trial
Ongoing/Registered Trials							
RENOBIOME (NCT07024238) [93]	Stable kidney transplant recipients (adults) *n* = 41	Sodium butyrate (oral) vs. placebo	1000 mg/day	12 weeks	Gut microbiome composition, serum and urinary metabolome profiling	Status: Enrolling by invitation	Randomized, double-blind, placebo-controlled clinical trial
METAKID (NCT06951581) [94]	Stable kidney transplant recipients, *n* = 41	SCFA oral formulation vs. placebo	200 mg/day	12 weeks + 12-week washout period	Serum/urine metabolomics; inflammatory and immunologic markers, eGFR/UACR, tacrolimus	Status: Primary data collection completed, results pending publication	Randomized, double-blind, placebo-controlled clinical trial
Pro-Kids (NCT05858437) [95]	Children/adolescents with CKD on hemodialysis (ages 12–20; *n* = 16)	Sodium propionate (oral) vs. placebo	2 × 500 mg	28 days	Tregs analysis, propionic acid serum levels, targeted metabolomics, fecal microbiome analysis, intestinal barrier function	Status: Recruiting	Multi-center, double-blind, randomized and placebo-controlled intervention study (pediatric)

The following table provides a summary of completed and ongoing clinical trials evaluating the therapeutic potential of short-chain fatty acids (SCFAs) in chronic kidney disease (CKD). **Abbreviations**: CKD, chronic kidney disease; HD, hemodialysis; T1D, type 1 diabetes; DKD, diabetic kidney disease; SCFA, short-chain fatty acid; Tregs, regulatory T cells; CRP, C-reactive protein; UACR, urinary albumin-to-creatinine ratio; eGFR, estimated glomerular filtration rate; HbA1c, glycated hemoglobin.

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
