# Peer review of "Gut Microbiome-Derived Short-Chain Fatty Acids in Glomerular Protection and Modulation of Chronic Kidney Disease Progression"

_nutrients, 2025, doi:10.3390/nu17172904_

Round 1
Reviewer 1 Report
Comments and Suggestions for Authors
Thank you for submitting your manuscript entitled ''Gut Microbiome-Derived Short-Chain Fatty Acids in Glomerular Protection and Modulation of Chronic Kidney Disease Progression''. After a careful evaluation, I would like to offer the following observations and suggestions to help strengthen your work and ensure its novelty and relevance to the field:
-
Lack of Novelty and Redundancy with Existing Literature
The manuscript currently presents a general overview of the topic without clearly demonstrating novel insights or a distinct contribution beyond existing reviews. Several publications (e.g., https://doi.org/10.2147/DDDT.S150825, https://doi.org/10.1016/j.biopha.2025.118214, https://doi.org/10.3390/toxins15090548, https://doi.org/10.3390/ijms23105354, etc.) have already provided comprehensive discussions on the role of SCFAs in kidney disease. To justify the need for this manuscript, I recommend clearly stating what new perspective, synthesis, or interpretation this review offers. -
Absence of a Clear Literature Gap
The manuscript would benefit from a more explicit identification of gaps in the current literature. Please articulate why an updated or revised review is necessary, particularly in relation to SCFA mechanisms in CKD pathophysiology, emerging clinical implications, or unresolved controversies. -
Missing Literature Selection Criteria
Even though the manuscript is not a critical review, the methodology behind the literature selection is not described. To enhance transparency and reproducibility, please include your inclusion/exclusion criteria, databases searched, time frames, and keywords used. -
Lack of Epidemiological Context
I recommend incorporating a brief section on the current global burden of CKD, including trends, risk factors, and regional disparities. This will help contextualize the significance of exploring novel therapeutic approaches such as SCFAs. -
Need for Clinical Evidence Summary
A table or dedicated sub-section summarizing existing clinical trials or intervention studies involving SCFA administration in CKD patients would significantly enhance the manuscript’s value. Key data may include patient demographics, intervention type, dosage, duration, measured outcomes, and conclusions.
I believe addressing these points will substantially improve the manuscript's scientific rigor and relevance.
Warm regards,
Author Response
REVIEWER 1
We sincerely appreciate the valuable and encourage comments. We have studied these comments carefully and have made corresponding corrections that we hope will meet with your approval. The changes in the revised manuscript are marked in red. The responses to the reviewers’ comments are provided below.
Comment 1: Lack of Novelty and Redundancy with Existing Literature
The manuscript currently presents a general overview of the topic without clearly demonstrating novel insights or a distinct contribution beyond existing reviews. Several publications (e.g., https://doi.org/10.2147/DDDT.S150825, https://doi.org/10.1016/j.biopha.2025.118214, https://doi.org/10.3390/toxins15090548, https://doi.org/10.3390/ijms23105354, etc.) have already provided comprehensive discussions on the role of SCFAs in kidney disease. To justify the need for this manuscript, I recommend clearly stating what new perspective, synthesis, or interpretation this review offers.
Response 1: We appreciate the reviewer’s comment and fully recognize the importance of distinguishing our work from existing reviews. The purpose of our manuscript is to present the protective role of SCFAs in glomerular injury during chronic kidney disease (CKD). Unlike most prior publications, which have addressed the kidney as a whole, our review is specifically focused on the glomerular compartment, with particular attention to podocyte biology. Given that our scientific expertise centers on glomerular filtration barrier and podocyte pathophysiology, we are well positioned to critically evaluate and synthesize the literature in this area. This background enables us to provide a perspective that bridges fundamental mechanisms with clinical relevance, offering insights into how SCFAs may contribute to the preservation of glomerular function. This emphasis is of particular significance, as glomerular integrity is essential for blood filtration and its dysfunction is a central driver of CKD progression. By concentrating on this critical structure, our review adds a focused and expert-driven contribution that complements and extends the scope of existing literature.
In addition, whereas many reviews combine data from both acute and chronic kidney injury, our work concentrates specifically on chronic kidney disease, offering a more in-depth and focused synthesis of evidence on long-term glomerular protection.
In summary, we believe that our manuscript offers a well-structured and comprehensive synthesis of the existing literature, reflecting both the depth of available data and our focused expertise in glomerular pathophysiology.
Comment 2: Absence of a Clear Literature Gap
The manuscript would benefit from a more explicit identification of gaps in the current literature. Please articulate why an updated or revised review is necessary, particularly in relation to SCFA mechanisms in CKD pathophysiology, emerging clinical implications, or unresolved controversies.
Response 2: We thank the reviewer for this insightful comment. We fully agree that explicitly identifying the gaps in the current literature is essential to demonstrate the value of our review. In our manuscript, the section entitled “Future Directions and Limitations” highlights several important areas where knowledge remains incomplete:
- Interindividual variability in gut microbiota composition, which underscores the need for more personalized diagnostic and therapeutic strategies.
- The necessity to extend early-phase clinical trials and address their current limitations, such as small sample sizes, lack of standardized and effective dosing regimens, suboptimal delivery methods, and insufficient data on long-term safety and efficacy of microbiome-targeted interventions in CKD management.
- The revised manuscript has been expanded to address translational gaps between animal and human studies, as well as the challenges associated with receptor targeting.
Moreover, section 4 entitled ‘Epigenetic Modifications in the Context of CKD and SCFAs’, highlights the need to further investigate specific molecular mechanisms, particularly the role of SCFAs in the selective inhibition of individual HDACs, which may provide novel insights into targeted therapeutic strategies.
By emphasizing these gaps, our review not only summarizes current evidence but also provides directions for future research and clinical translation.
Comment 3: Missing Literature Selection Criteria
Even though the manuscript is not a critical review, the methodology behind the literature selection is not described. To enhance transparency and reproducibility, please include your inclusion/exclusion criteria, databases searched, time frames, and keywords used.
Response 3: According to the suggestion, we have added a Methods section that clearly describes the literature search strategy, including the inclusion criteria, databases searched, time frames, and keywords used.
Comment 4: Lack of Epidemiological Context
I recommend incorporating a brief section on the current global burden of CKD, including trends, risk factors, and regional disparities. This will help contextualize the significance of exploring novel therapeutic approaches such as SCFAs
Response 4: We have incorporated a concise fragment with key epidemiological data on CKD into the Introduction section.
Comment 5: Need for Clinical Evidence Summary
A table or dedicated sub-section summarizing existing clinical trials or intervention studies involving SCFA administration in CKD patients would significantly enhance the manuscript’s value. Key data may include patient demographics, intervention type, dosage, duration, measured outcomes, and conclusions.
Response 5: We thank the reviewer for this valuable suggestion. We have added a dedicated subsection, Clinical evidence for SCFA supplementation in CKD together with summary tables that compile patient demographics, intervention type, dose, duration, outcomes, and main conclusions from human interventional studies. To provide greater clarity, we distinguished between completed trials and those that are still ongoing.

Reviewer 2 Report
Comments and Suggestions for Authors
Thank you very much for the opportunity to revise this interesting literature review entitled "Gut Microbiome-Derived Short-Chain Fatty Acids in Glomerular Protection and Modulation of Chronic Kidney Disease Progression". It is a highly relevant subject—the gut-kidney axis and SCFAs’ protective effects in CKD—which is of increasing interest in nephrology and microbiome research. However, it will improve with some corrections:
1-Several points are repeated in different sections (e.g., butyrate effects, fiber-SCFA link). The manuscript will improve if redundancy or untidiness are avoided.
2-Consider shortening some sections which are currently too long, for example the sections 5.3-5.5.
3-In general the English is ok, but some sentences are long or overly complex. For example: "This dual approach helps prevent further damage to kidney function by limiting toxin exposure and reducing proteinuria, key factors in preserving long-term kidney health and mitigating metabolic disease progression".
4-A more critical view of limitations (e.g., differences in human vs. animal data, SCFA variability, challenges in receptor targeting) would be necessary to add objectivity.
5-Figures could be improved, I suggest the authors to look for more professional components to carry out the diagrams. For example in https://smart.servier.com/
6- Effective doses, routes of administration, or bioavailability of SCFAs in humans could be further explored.
Author Response
REVIEWER 2
We sincerely appreciate the valuable and encourage comments. We have studied these comments carefully and have made corresponding corrections that we hope will meet with your approval. The changes in the revised manuscript are marked in red. The responses to the reviewers’ comments are provided below.
Comment 1: Several points are repeated in different sections (e.g., butyrate effects, fiber-SCFA link). The manuscript will improve if redundancy or untidiness are avoided.
Response 1: We appreciate this valuable comment. We intentionally separated the discussion of SCFAs and dietary fiber into distinct subsections to underscore the growing importance of dietary fiber while enhancing the manuscript’s readability and clarity. However, in line with the reviewer’s suggestion, we carefully re-evaluated the manuscript for potential redundancies and removed repetitive content. For instance, in the subsection Role of Dietary Fiber in Oxidative Stress in CKD, we deleted the fragment concerning propionate.
Comment 2: Consider shortening some sections which are currently too long, for example the sections 5.3-5.5.
Response 2: We thank the reviewer for this comment. In the revised manuscript, we have shortened several sections to improve clarity and conciseness.
Comment 3: In general the English is ok, but some sentences are long or overly complex. For example: "This dual approach helps prevent further damage to kidney function by limiting toxin exposure and reducing proteinuria, key factors in preserving long-term kidney health and mitigating metabolic disease progression
Response 3: Thank you for pointing this out. Our manuscript was indeed thoroughly proofread by a professional native English-speaking editor with expertise in the cellular and molecular mechanisms of diabetes and diabetic nephropathy prior to submission. In addition, we have shortened the sentence in question.
Comment 4: A more critical view of limitations (e.g., differences in human vs. animal data, SCFA variability, challenges in receptor targeting) would be necessary to add objectivity.
Response 4: We thank for this insightful comment. In the revised version, we have expanded the subsection Future Directions and Limitations to include a more critical discussion of the suggested key challenges.
Comment 5: Figures could be improved, I suggest the authors to look for more professional components to carry out the diagrams. For example in https://smart.servier.com/
Response 5: Figure 3 was revised to achieve a more professional appearance using resources from smart.servier.com. As it also serves as the graphical abstract, this figure plays a key role in the manuscript. In contrast, Figures 1 and 2 were left unchanged because we did not find sufficient components in this program to enable their improvement.
Comment 6: Effective doses, routes of administration, or bioavailability of SCFAs in humans could be further explored.
Response 6: We have added a dedicated subsection, Clinical evidence for SCFA supplementation in CKD, accompanied by summary table that synthesize patient demographics, route, dose, duration, and key outcomes from human intervention

Round 2
Reviewer 1 Report
Comments and Suggestions for Authors
-please mention the platforms used for literature selection
-SCFAs are also known as postbiotics, so please refer to them in your manuscript, especially in section 2
-Table 1 looks very good
Author Response
Comments 1: please mention the platforms used for literature selection.
Response 1 : Our search was conducted using PubMed, as indicated in the description
Comments 2: SCFAs are also known as postbiotics, so please refer to them in your manuscript, especially in section 2
Response 2: In line with the suggestion, we have incorporated information regarding the classification of SCFAs as postbiotics.
Comments 3: Table 1 looks very good.
Response 3: We thank the reviewer for the positive assessment of Table 1.
